# Dads at Mealtimes: Associations between Food Security, Household and Work Chaos, and Paternal Feeding Practices among Australian Fathers Living with Disadvantage

**DOI:** 10.3390/nu16020205

**Published:** 2024-01-08

**Authors:** Jeffrey T. H. So, Smita Nambiar, Rebecca Byrne, Danielle Gallegos, Kimberley A. Baxter

**Affiliations:** 1Centre for Childhood Nutrition Research, Faculty of Health, Queensland University of Technology, 62 Graham Street, South Brisbane, QLD 4101, Australia; smita.nambiar@qut.edu.au (S.N.); ra.byrne@qut.edu.au (R.B.); danielle.gallegos@qut.edu.au (D.G.); kimberley.baxter@qut.edu.au (K.A.B.); 2School of Exercise and Nutrition Sciences, Faculty of Health, Queensland University of Technology, Victoria Park Road, Kelvin Grove, QLD 4059, Australia

**Keywords:** fathers, feeding practices, responsive feeding, food parenting, food security, household chaos, paternal behavior, family meals

## Abstract

Understanding how fathers engage in feeding while experiencing disadvantage is important for family-focused interventions. A cross-sectional online survey involving 264 Australian fathers was conducted to explore feeding involvement and the relationships between feeding practices, food insecurity, and household and work chaos. Practices related to coercive control, structure, and autonomy support were measured for two age groups (<2 years and 2–5 years). Multivariable linear regression was used to examine the associations for each practice. Three-quarters of the sample were food insecure, impacting adults more than children, and correlated with household chaos. Food insecurity was associated with increased ‘persuasive feeding’ and ‘parent-led feeding’ in younger children. Household chaos was positively associated with coercive control practices in both younger and older child groups, with the strongest associations for ‘using food to calm’ and ‘overt restriction’, respectively. In older child groups, household chaos was negatively associated with ‘offer new foods’ and ‘repeated presentation of new foods’. Structure practices had no significant relationships with any factors, and work chaos did not predict any feeding practices. These findings emphasize a need for societal and structural support to address food insecurity and household chaos. Tailored strategies are crucial to support fathers in responsive feeding.

## 1. Introduction

The Nurturing Care Framework proposes that children need good health, adequate nutrition, safety and security, learning opportunities, and responsive caregiving for optimal early childhood development [1]. Optimal nutrition is key to supporting children’s physical and cognitive development and maintaining a healthy weight [2]. However, in Australia, only 4% of children meet fruit and vegetable recommendations [2], and more than one-fifth of children consume sugar-sweetened beverages at least once a week [3]. Children living with disadvantage in their first year of school are also three times more likely to be vulnerable across two or more developmental domains, particularly in the areas of physical health and wellbeing, and language and cognitive skills [4]. In the early years, parents, including fathers, play an integral role in providing nurturing care and feeding of their children. Feeding and caregiving are inextricably connected, and parental feeding is central to parent–child interactions [5].

Feeding practices refer to goal-oriented strategies that parents use during child feeding. These practices fall into three overarching constructs: ‘structure’, ‘autonomy support’, and ‘coercive control’ [6]. ‘Structure’ involves setting routines and role modeling. ‘Autonomy support’ includes reasoning and encouragement. Feeding strategies that fall under these constructs are known as responsive feeding practices, characterized by prompt, emotionally supportive, contingent, and developmentally appropriate reciprocity between the child and their caregiver [7,8]. Responsive feeding is associated with improved diet quality and eating behaviors [9]. ‘Coercive control’ describes parental dominance over children’s eating, such as pressuring to eat or using food as a reward [6]. These practices are non-responsive and override a child’s innate ability to self-regulate their appetite, leading to eating for reasons other than hunger [7,9].

Child feeding research has predominantly focused on mothers [10,11]. For example, in child obesity prevention and treatment trials, fathers constituted only 6% of cases when one parent was involved (n = 80) [12]. This underrepresentation of fathers potentially hinders the development of family-focused feeding interventions and misses the opportunity to engage men in nurturing care. Emerging studies suggest that fathers make a distinct contribution to the family food environment [13]. Fathers in Australia, the United States (US), and Denmark report high involvement in family food work, including feeding children [14,15,16]. Some studies suggest fathers adopt greater levels of coercive control and lower levels of structure and autonomy support practices than mothers [15,17]. The existing literature highlights the independent effects of paternal practices on children’s eating and weight outcomes [10,17]. This influence marks the importance of fathers, particularly when considering the evolving dynamics of family structures and shifting societal gender roles.

Responsive feeding depends on the caregiver providing structure and routine [8]. Disadvantage, characterized by financial hardship and social exclusion, can disrupt the household, leading to poorer family and child outcomes [18]. Food insecurity is a parameter of disadvantage that can introduce instability and dysfunction into a household. Food insecurity occurs when people have inadequate access to food of sufficient quantity and quality to meet dietary needs and preferences [19]. It affects approximately 8% of people in high-income countries and is more prevalent in disadvantaged areas [20,21]. Studies among mothers indicate that food insecurity leads to less responsive feeding and more coercive control practices [22,23]. Financial barriers may limit access to perishable and nutritious foods, impacting repeated food exposure crucial for developing food acceptance in children [24]. Food insecure households may also experience time constraints, greater stress, and family conflict [25], reducing the frequency of family meals [26]. Another factor contributing to instability is chaos in the home and work environments. Household chaos, characterized by a lack of organization or environmental confusion [27], has been associated with adverse family and child outcomes and appears to reduce parental responsiveness [28]. Work chaos, marked by work-related stress and inflexible schedules, can affect fathers in employment [29], potentially influencing paternal feeding practices.

Understanding paternal feeding involvement and practices within the context of disadvantage can provide valuable insights for child nutrition research and intervention design. This study aimed to achieve three objectives among Australian fathers: (1) describe paternal involvement in child eating, (2) assess the prevalence and severity of food insecurity while exploring its relationship with household and work chaos, and (3) identify paternal feeding practices and examine how these practices are associated with household food insecurity, household and work chaos, and other sociodemographic factors.

## 2. Materials and Methods

### 2.1. Study Procedure and Sample

This cross-sectional study recruited Australian fathers experiencing disadvantage. Participation was promoted through three methods: (1) paid social media advertisements targeting males interested in parenting, fatherhood, and food; (2) study information posted on relevant Facebook pages; (3) study flyers shared with organizations providing family and child services and food relief services. All recruitment materials were tailored to engage fathers. A sample size of 200 was considered acceptable after consultation with a statistician. This study was approved by the Queensland University of Technology Human Research Ethics Committee (HREA 2022-5253-7746).

Participants self-screened using two criteria: (1) being a father or male caregiver of a child aged six months to five years and (2) affirmatively responding to the question ‘Do you sometimes struggle to pay the bills?’ Additional eligibility criteria included being at least 18 years of age, English proficiency, and the absence of health conditions that affected appetite, feeding, and growth in the index child. Participants with multiple children in the specified age range were asked to respond for the child with whom they had more feeding involvement. The questionnaire was hosted on Research Electronic Data Capture (REDCap, https://projectredcap.org/resources/citations/ accessed on 4 January 2024) [30,31] and took approximately 20 min to complete. After survey completion, participants could enter a prize draw to win one of four AUD 100 gift cards. Figure 1 displays the participant flow diagram.

### 2.2. Measures

#### 2.2.1. Sociodemographic Data

Participants reported their age, height, weight, employment status, education level, relationship status, postcode of primary residence, household income, source of income, housing type, cultural/ethnic background, country of birth, and household composition. The frequency of residential moves in the past 12 months was collected as a proxy of household stability. Information about the index child, including their age, gender, living arrangements, participant–child relationship, and childcare attendance, were collected. Fathers who did not reside with the child full-time reported the number of days they lived together in an average fortnight. Participants also reported their stress level and ability to manage stress using two items [26].

#### 2.2.2. Involvement

Paternal involvement questions were adopted from the Early Childhood Longitudinal Study [16]. Fathers reported their influence on major decisions about their child’s nutrition (range from ‘no influence’ to ‘a great deal of influence’) and the frequency with which they prepared meals for their child and assisted them with eating (feeding or eating with the child) in the past month (range from ‘not at all’ to ‘more than once a day’).

#### 2.2.3. Early Child Feeding and Paternal Feeding Practices

Fathers were asked to report on breastfeeding practices and the introduction of complementary foods. Parental feeding practices were assessed using three validated questionnaires [32,33,34]. The Feeding Practices and Structure Questionnaire solid feeding version (FPSQ-S) was administered to children under the age of 2 years and the 28-item version (FPSQ-28) was administered to children over 2 years old. The FPSQ-S consists of six constructs reflecting non-responsiveness/coercive control (‘persuasive feeding’, ‘parent-led feeding’, ‘using food to calm’, and ‘using (non-) food rewards’) and structure (‘feeding on demand’ and ‘family meal environment’), where ‘family meal environment’ and ‘using (non-food) rewards’ were applicable only for children 12 months or older [32]. The FPSQ-28 consisted of seven constructs and a single item indicator for family meal setting [33]. Four constructs related to coercive control included: ‘persuasive feeding’, ‘reward for eating’, ‘reward for behaviour’, and ‘overt restriction’. Three constructs related to structure included ‘covert restriction’, ‘structure meal timing’, and ‘structure meal setting’. A domain from the Food Parenting Inventory (FPI) was administered to fathers with children over 2 years old to assess autonomy support practices and encompassed four constructs: ‘encouraging exploration of new foods’, ‘offer new foods’, ‘urging the child to eat new foods’, and ‘repeated presentation of new foods’ [34]. Participants responded using a 5-point Likert scale, with a ‘non applicable’ response available for FPSQ-S items.

#### 2.2.4. Food Security Status

Household food security status was assessed using two measures. Firstly, a single-item question from the Australian National Health Survey (NHS) was included to enable comparison to nationally available data [35]. Secondly, the 18-item US Department of Agriculture Household Food Security Survey Module (HFSSM) with adult and child-specific indicators was administered [36].

#### 2.2.5. Household and Work Chaos

Household chaos was measured using the six-item version of the Confusion, Hubbub, and Order Scale (CHAOS) [37]. Participants described their home environment by responding to statements such as ‘It’s a real zoo in our home’ on a 5-point Likert scale, ranging from ‘definitely untrue’ to ‘definitely true’. Work chaos was assessed with a 4-item measure, including questions such as ‘My shift and work schedule cause extra stress for me and my child’ [29]. This was administered only to participants who were employed or in an apprenticeship. Response options ranged from ‘never’ to ‘always’.

### 2.3. Data Analysis

Data analyses were completed using the Statistical Package for the Social Sciences version 29 (IBM SPSS Statistics) [38]. A parental stress index was created by dividing the overall stress score by the management of the stress score. The index was dichotomized into ‘well-managed stress’ (<1) and ‘unmanaged stress’ (≥1) [26]. Equivalized household income was calculated by dividing the midpoint of the household income bracket by an equivalence factor [39]. Body mass index (BMI) was calculated based on self-reported weight and height. Mean scores were calculated for each feeding construct. Higher scores indicated more endorsement of the practices, except for ‘feeding on demand’, where a high score indicated adherence to a feeding routine. HFSSM items were coded according to the guide, with missing values replaced using the direct imputation method [36]. Affirmative responses were summed to generate a raw score, allowing the categorization of the severity of food insecurity at the household, adult, and child levels. Household food security status was also dichotomized to ‘food secure’ (high and marginal food security) or ‘food insecure’ (low and very low food security) for data analysis. CHAOS items were summed to provide a total score ranging from 6 to 30, with a higher score indicating a more chaotic home environment. Each work chaos item was coded and summed to generate a total score between zero and four, where a higher score indicated greater work chaos. For participants who reported being unemployed, on parental leave, student, or unable to work, a score of 0 was imputed to maximize the sample size. Household chaos and work chaos were treated as continuous variables.

Preliminary analyses assessed variable distribution, normality, and missing data. Cronbach’s alphas were calculated to verify the psychometric properties of the feeding instruments. Most feeding constructs showed acceptable Cronbach’s alphas (>0.6), ranging from 0.68 to 0.94. However, ‘feeding on demand’ (α = 0.55), ‘structured meal timing’ (α = 0.39), and ‘urge child to eat new food’ (α = 0.58) did not meet the acceptable threshold (<0.6) and were excluded from the regression analysis (See Appendix A). Issues with response distribution and normality (visually on histogram, high kurtosis/skewness values > +/−3) were noted. The constructs ‘family meal environment’ and ‘using (non-) food reward’ were applicable to fathers of children 12 to 24 months (n = 56). Missing values for the remaining feeding constructs ranged from 9% to 16%, with the highest being ‘family meal setting’. Since missing values appeared to align with the order of the feeding constructs in the survey (suggesting they were missing, not at random), a complete case analysis was conducted.

Descriptive statistics were used to describe father and child sociodemographic characteristics, feeding involvement and practices, family meal setting, food security status, and CHAOS and work chaos scores. Independent *t*-tests and one-way ANOVA tests were used to compare the mean differences in CHAOS and work chaos scores for each food security status. For each feeding practice, Pearson or Spearman correlations were used to assess associations with three continuous sociodemographic variables: equivalized household income, paternal age, and child age. Six variables were dummy-coded and tested using an independent *t*-test or Mann–Whitney test: education level, number of residential moves, number of children, child gender, paternal BMI, and stress.

Bivariate analyses (Fisher’s exact test and one-way ANOVA) were conducted to examine the independent associations between family meal setting, food security status, household chaos, and work chaos scores within the older child group. Multiple linear regression was employed to explore the independent associations between key variables, feeding practices, and covariates for younger and older child groups. Six models were constructed to examine the individual and combined effects of food security, household and work chaos, and other covariates on feeding practices. Purposeful selection was used to produce the final model. All sociodemographic variables with *p* < 0.2 in the univariate analyses were initially included, and then non-significant variables (*p* > 0.05) were removed in subsequent steps, with attention to changes in coefficients exceeding 20%. Any eliminated variables with confounding effects were reintroduced, guided by existing knowledge. The assumptions for regression models were assessed, including independence of observations, linearity, homoscedasticity of residuals, multicollinearity, and residual normality. ‘Using (non-) food rewards’ was excluded due to serious heteroscedasticity and violations of residual normality. The significance level was set at *p* < 0.05.

## 3. Results

### 3.1. Participants’ Characteristics

The online survey was accessed by 736 participants from March to September 2022, and 314 commenced the survey. The final analytic sample included 264 participants (younger child group (<2 years; n = 105) and older child group (2–5 years; n = 159)). Two participants had children slightly outside the study age range (5.3 and 5.4 months) but were included in the analysis as their children had started consuming solid foods. Sociodemographic characteristics of fathers and children are presented in Table 1.

### 3.2. Paternal Involvement in Child Eating

All children had commenced complementary feeding at the time the questionnaire was completed. Almost all fathers (98%) felt they had at least some influence in making decisions about their child’s nutrition, with the majority involved in preparing meals (67%) and assisting their child with eating (69%) at least once a day or more than once a day. Table 2 provides the descriptive analysis of paternal feeding involvement.

### 3.3. Food Security Status and Household and Work Chaos

#### 3.3.1. Food Security Status

Among the 222 participants who completed the food security measures, 37% of households were classified as food insecure using the NHS single item. However, with HFSSM, 77% of households were classified as food insecure. The prevalence and severity of food insecurity in the total sample, among adults and children, are summarized in Table 3. Detailed responses to HFSSM individual items and food security status for both child age groups are provided (See Appendix A).

#### 3.3.2. Associations with Household and Work Chaos

The mean raw score for HFSSM in the sample was 6.48 (SD = 4.23, n = 222). For household chaos and work chaos, the mean scores were 16.03 (SD = 4.45, n = 220) and 1.27 (SD = 1.30, n = 214), respectively. Table 4 compares CHAOS and work chaos scores by the severity of household food insecurity. Food insecure households had a significantly higher mean CHAOS score than food secure households (16.42 ± 4.42 versus 14.72 ± 4.35; *p* = 0.017). A graded effect was evident, whereby the CHAOS score increased with the severity of household food security. However, this trend was only significant in the older child group (*p* = 0.01). Food security status was not significantly associated with work chaos score (*p* = 0.76).

### 3.4. Paternal Feeding Practices and Their Associations with Food Security and Household and Work Chaos

#### 3.4.1. Paternal Feeding Practices

Descriptive statistics for paternal feeding practices are provided in Appendix A. Practices related to rewards and using food to calm had the lowest means overall (using (non-) food rewards: 1.9 (younger child group) and reward for eating: 2.5 (older child group)). For the younger child group, structure practices like feeding on demand and family meal environment displayed higher means (3.5 and 3.9, respectively) compared with coercive control practices (range: 1.9–2.8). Amongst the older child group, the highest mean scores for each coercive control, structure, and autonomy support construct were overt restriction (3.7), structured meal setting (3.7), and encourage exploration of new foods (3.9), respectively. In contrast, the lowest for each construct were reward for eating (2.5), covert restriction (2.8), and repeated presentation of new foods (3.4), respectively.

#### 3.4.2. Family Meal Setting and Food Security and Household and Work Chaos

When examining family meal settings in the older child group (n = 134), most fathers indicated that their children always (32%) or often (40%) consumed the same meals as the rest of the family. There were no significant associations between family meal setting categories and food security (*p* = 0.755), mean CHAOS score (F_4,128_= 1.58, *p* = 0.183), or work chaos score (F_4,128_= 0.892, *p* = 0.471) (see Appendix A).

#### 3.4.3. Multivariable Regression Predicting Feeding Practices

The associations between feeding practices and household food insecurity and household and work chaos after adjusting for covariates are presented in Table 5. Other models examining individual and combining effects of the key variables are provided in Appendix A.

In the younger child group, three coercive control (using food to calm, persuasive feeding, and parent-led feeding) and one structure (family meal environment) feeding practices were included in regression analyses. Using food to calm was only positively associated with household chaos after adjusting for food insecurity, work chaos, child gender, and income (*p* < 0.001). Persuasive feeding was positively associated with food insecurity (*p* = 0.016) and household chaos (*p* < 0.014). Parent-led feeding was positively associated with food insecurity (*p* = 0.030) and paternal education. The final model for family meal environment was not statistically significant, although there appeared to be a relationship with child gender; when compared with girls, the family meal environment score was 0.570 units lower in boys (*p* =0.043).

Among the older child group, four coercive control (reward for behavior, reward for eating, persuasive feeding, and overt restriction), two structure (covert restriction and structured meal setting), and three autonomy support (offering new foods, repeated presentation of new foods, and exploration of new foods) feeding practices were included in regression analyses.

In the final model, reward for behavior was associated with household chaos (*p* = 0.025) and fathers’ age (*p* = 0.006). Furthermore, reward for eating was positively associated with household chaos (*p* = 0.033), residential moving in the last year (*p* = 0.01), and child age (*p* = 0.002). Unlike the younger age group, persuasive feeding was not associated with food insecurity or household chaos, nor was it associated with work chaos; it was associated with residential moving in the last year (*p* = 0.005), fathers’ age (*p* = 0.043), and education (*p* = 0.016). Overt restriction was associated with household chaos (*p* = 0.005) and higher BMI (*p* = 0.010). The models for covert restriction and structured meal settings were not significant in this sample. Offering new foods was negatively associated with household chaos in the final model after adjusting for education (*p* = 0.039); however, it was not associated with food security and work chaos (both *p* > 0.05). Exploration of new foods were found to have no association with food security status, household chaos, or work chaos (all *p* > 0.05) after controlling for education and child’s age. However, household chaos was negatively associated with repeated presentation of new foods after adjusting for education and child’s age (*p* = 0.013).

## 4. Discussion

This study examined paternal involvement in child feeding and the prevalence of food insecurity among Australian fathers experiencing disadvantage. It assessed paternal feeding practices across two child age groups, shedding light on the relationship between household food security, household and work chaos, and feeding practices. The novel findings from this research highlight that food insecurity and household chaos were associated with paternal coercive control and autonomy support practices. This has implications for future research and practices promoting responsive feeding, an important aspect of adequate nutrition and responsive caregiving within the nurturing care framework [1].

Approximately 65–70% of fathers reported active involvement in daily family meal preparation and child feeding. This corresponds with recent research on Australian fathers with children under five, where approximately 75% assumed equal or greater responsibilities than their partners in meal planning, food shopping, and cooking [14]. Paternal involvement was higher than previously reported a decade ago, when just 42% of Australian fathers indicated high involvement in organizing meals [41]. This could be due to changing work flexibility, such as work-from-home arrangements and the growing employment trend of women, contributing to more shared responsibilities within the household. In a US study, 43.2% of fathers with 2-year-old children reported having a great deal of influence regarding their child’s nutrition [16], while the current study found an even higher proportion at 57%. The increased involvement may have been influenced by the COVID-19 pandemic, as the survey was distributed when Australia was emerging from pandemic-related restrictions, potentially affecting fathers’ time spent at home and childrearing responsibilities. However, Kuswara and colleagues noted that COVID-19 restrictions did not significantly impact fathers’ responsibility for food work [14]. While mothers continue to bear the primary responsibility for child feeding [13,15], the increasing and sustained paternal involvement, even in times of financial hardship, offers an opportunity for early feeding interventions involving fathers.

The prevalence of food insecurity among fathers living with disadvantage is not well-reported. Three-quarters of study participants were classified as food insecure through the HFSSM. Comparisons between populations were challenging due to the varied measurement tools [42] and the primary responder typically being the female head of household [25,42]. Food insecurity rates were not consistently separated for fathers when both parents were included [43,44]. When comparing with the NHS measure, 37% of the sample were food insecure—a sevenfold increase from the pre-COVID-19 national average of 5% [35] and 4.5 times greater than the prevalence among mother–father pairs with young children residing in disadvantaged neighborhoods (8%) [44]. This single-item measure has been shown to underestimate food insecurity compared with more comprehensive tools [45], as illustrated in this study (37% versus 77%). Using the HFSSM, an Australian study revealed an 85% prevalence of food insecurity among male participants experiencing entrenched disadvantage (i.e., meeting two or more of the following criteria: under- or unemployment, unstable housing, disability, mental disorders, inadequate social support, low education, or reliance on welfare payments) [46]. Sixty-two percent of all adults experienced very low food security [46] compared with almost half of the adults in the current study. This implies that three in five adults were reducing the size of meals or skipping meals, with over half having to go hungry. Food insecurity had a more profound impact on adults, with a higher percentage experiencing very low food security compared with children (49% versus 2%). These results suggest that adults prioritized their child’s food intake, aligning with a pattern reported in the literature among mothers [47]. Food insecurity can lead to adverse health and social implications for both adults and children, such as nutrient inadequacies [48,49] and poor health [50], as well as poorer developmental and behavioral outcomes in children [21]. Food-insecure households reported higher household chaos than those who were food secure, serving as an additional factor that can disrupt daily routines. The disproportionately high rate of food insecurity within this disadvantaged cohort presents significant implications for public health and social policy.

Consistent with studies on mothers and fathers [32,51,52], fathers in the current study generally endorsed structure and autonomy support feeding practices. For example, 70% of fathers often or always established family meals where children ate the same meals as the rest of the family. Offering new foods and encouraging exploration of new foods was highly endorsed among fathers with an older child (mean: 3.7–3.9). Fathers of young children also demonstrated ‘feeding on demand’ scores (mean: 3.5), representing more adherence to a feeding schedule. However, it remains unclear at which developmental age children would benefit more from structure, such as setting a meal routine instead of feeding on demand where the infant determines the timing of feeds. In relation to coercive control practices, fathers typically refrained from using rewards for eating or foods to calm. This is favorable, as practices like bribing a child to eat healthy foods have been shown to decrease the child’s liking for the target food. In contrast, practices like persuasive feeding (parental use of pressure) and overt restriction (limiting food access that children notice) became more prevalent among fathers with older children. Coercive control practices were adopted more commonly in fathers than in mothers in previous studies [53,54,55]. Considering the trend of these practices as children age and their potential impact on child eating and self-regulation capacity, this study underscores the significance of early feeding interventions targeting fathers with infants.

The findings provide some support that food insecurity may elevate the adoption of coercive control practices, especially among the young child group. It echoes some studies conducted with mothers that report higher use of restriction and pressure to eat behaviors [23,56,57], while contradicting others [43,58]. The anxiety around managing financial resources and accessing sufficient food may lead parents to restrict children’s food intake during shortages [59] or to pressure them to eat the available foods [22]. This is concerning, as these feeding practices may act as pathways through which food insecurity affects child nutrition and health [25]. Additionally, the lack of a relationship between food insecurity and practices related to structure and autonomy support in this study mirrors previous findings among low-income mothers [43]. A possible explanation for these results could be the role of parental coping mechanisms in increasing parents’ capacity to provide responsive feeding. For example, responsive feeding practices were adopted by parents (predominately mothers) with high food resource management skills regardless of food security status [60]. It can be speculated that fathers in this study may develop financial and food-based strategies to cope with limited resources. More research is needed to investigate the potential role of paternal resource management skills in mitigating the deleterious effects of food insecurity on feeding practices. McCurdy and colleagues suggest food insecurity may influence child feeding through other processes, such as parental or child stress [43]. In the study, parental stress and depressed mood reported earlier in the day predicted less homemade foods and more packaged meals in the evening within food insecure households but not food secure families. Though parental stress was considered in the current study, a two-item measure was unlikely to capture the fluctuating nature of stress associated with cycles of food insecurity. The HFSSM assessed parents’ food situation in the last 12 months, which may not reflect the immediate consequences of food insecurity, including day-to-day variations. As chronic food insecurity has been shown to aggravate non-responsive feeding practices [56], longitudinal research can help understand how transient or chronic food insecurity and parental skills influence feeding across child development.

The pandemic brought unprecedented change in families’ lives; parents spent more time at home, resulting in increased family meals together [56]. The absence of associations between food insecurity and structure practices such as family meals and autonomy support practices like repeated presentation of new foods suggests that these practices may have become more common amongst fathers. Indeed, COVID-19-specific stress, such as concerns around jobs and necessities, led to greater efforts to establish meal routines and increased engagement with children during mealtimes [61]. This is beneficial, since family meals and food exposures are linked to improved child nutritional intake [62]. However, families respond to the experience of food insecurity and other stressful environmental factors differently. It is increasingly recognized that marginal food security, characterized by anxiety over food shortage, can affect parenting [63], thus potentially altering feeding practices. However, this category was classified as food secure according to the HFSSM guide. The relationship between food insecurity severity and paternal feeding should be investigated further with a larger sample.

The observation that household chaos predicted several coercive control and autonomy support feeding practices while work chaos did not, underscores the significance of disorganization within the home environment. Research on the impact of chaos on feeding practices is scant. A previous study found that chaos was associated with mothers using food to soothe, possibly to alleviate their own and children’s emotional distress [64]. Household chaos has also been reported to decrease family meal frequency and increase barriers to meal preparation (e.g., lack of time or energy to plan and cook meals) [26]. Surprisingly, this phenomenon was not observed with structure practices in the current study. This might be attributed to the high proportion of fathers reporting frequent family meals or differences in how they interpret feeding items compared with mothers, especially when their roles appeared to be assisting mothers in family meals [65]. Household chaos has been linked to children’s unhealthy food-related behaviors, health, and weight outcomes [66,67]. The factors that contribute to household chaos and the mechanisms through which it interferes with parental responsive practices should be investigated further.

This study sheds light on paternal involvement and practices among Australian fathers living with disadvantage. The effective use of a subjective screening criterion for financial hardship, developed after consultation with families and organizations, proved effective compared with other traditional measures like income for identifying families from lower socioeconomic backgrounds. The inclusion of the HFSSM, a comprehensive and validated measure of food insecurity, and measures of household and work chaos represent a strength in paternal feeding studies, which is seldom investigated. These contextual factors can guide future interventions, offering insights for tailored feeding advice to families experiencing disadvantage.

This study has limitations that should be considered when interpreting the results. The high proportion of food insecurity among fathers challenged investigations of linear relationships, potentially contributing to null findings in some practices. A larger sample size is essential for robust conclusions. Data collection occurred during the later stages of the COVID-19 pandemic (March to September 2022), and aspects such as pandemic-related changes in work/childcare arrangements and food access were not explored. Utilizing the short form version of CHAOS reduced participant burden, but its limited validation posed constraints. Work chaos, a relatively understudied aspect of families’ social context, was imputed for fathers who were unemployed to retain sample power. Further research could explore other sources of chaos (e.g., family conflict) and their impact on paternal feeding by employing various methods, including observations. Normality violation was observed for the ‘use of (non-) food rewards’ construct in the younger child group, represented by the low reported use of this practice. Low internal consistency for three feeding constructs, consistent with original scale development studies [32,33], posed another limitation. Existing measurement tools are seldom validated explicitly for fathers or disadvantaged samples [17]. Validation efforts, which might include cognitive interviews and comparisons with direct observations, are crucial. Selection bias and social desirability bias in self-reported questionnaires may exist, leading fathers involved and interested in feeding to participate or report more favorable practices. Despite attempts to recruit diverse male caregivers, non-biological or non-residential fathers are underrepresented, necessitating specific attention in future feeding studies. Lastly, the cross-sectional study design was unable to determine the directionality of relationships.

## 5. Conclusions

Fathers in this study demonstrated notable involvement in child feeding but faced challenges related to food insecurity and household chaos, impacting paternal feeding practices across development stages. These findings provide insight for designing interventions that support fathers with responsive feeding. Researchers and practitioners should integrate strategies considering fathers’ experiences of socioeconomic disadvantage to mitigate their impact on child feeding. From a policy perspective, ensuring stable financial and social access to nutritious foods and addressing household chaos is crucial for promoting family well-being. This contributes to building fathers’ capacities to provide nurturing care for optimal child nutrition and development.

## Figures and Tables

**Figure 1 nutrients-16-00205-f001:**
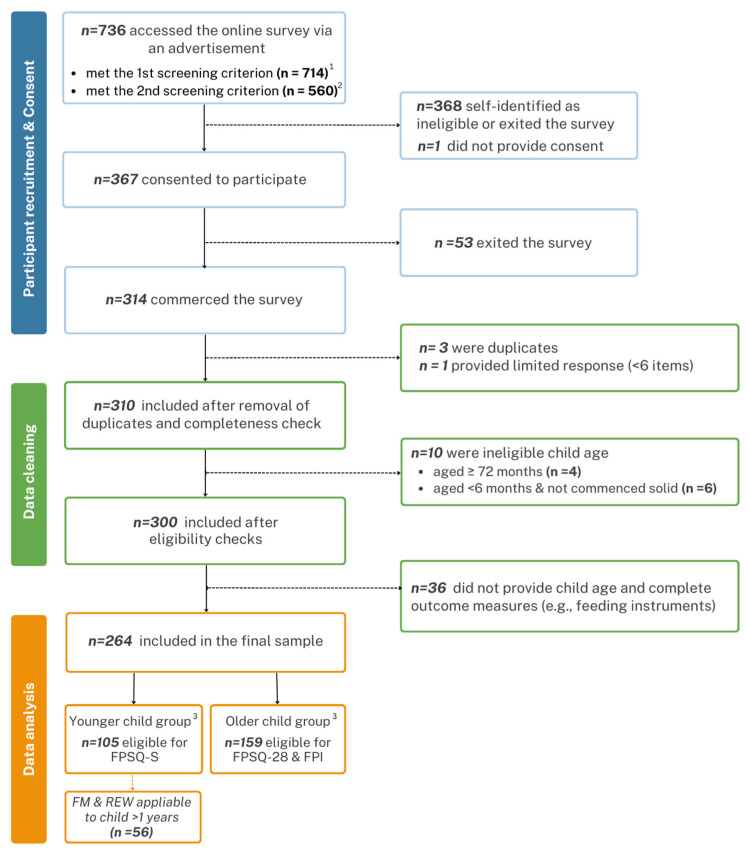
Participant flow diagram including the final sample size and reasons for exclusion. FPSQ-S—Feeding Practices and Structure Questionnaire (solid version); FPSQ-28—Feeding Practices and Structure Questionnaire-28; FPI—Food Parenting Inventory; FM—family meal environment; REW—using (non-food) rewards. ^1^ Criterion was ‘Are you a father or male caregiver with a child aged 6 months to 5 years?’ ^2^ Criterion was ‘Do you sometimes struggle to pay the bills?’ ^3^ Younger child group: <2 years; older child group: 2–5 years.

**Table 1 nutrients-16-00205-t001:** Sociodemographic characteristics of fathers and children.

Demographics	Total Sample (n = 264)	Younger Child Group (<2 Years) (n = 105)	Older Child Group (2–5 Years) (n = 159)	Missing Total, n (%)
** *Father* **				
**Age in years,** **Median (IQR)**	34.0 (30–37)	32.0 (29–35)	35.0 (30–38)	2 (0.8)
**BMI ^1^ category, n (%)**				3 (1)
<25	72 (27)	30 (29)	42 (26)	
≥25	189 (72)	73 (70)	116(73)	
**Highest level of education, n (%)**				0
Non-university education	166 (63)	63 (60)	103 (65)	
University education	96 (36)	41 (39)	55 (35)	
Prefer not to say	2 (1)	1 (1)	1 (1)	
**Marital status, n (%)**				0
Married/De facto	221 (84)	99 (94)	122 (77)	
Divorced/Separated	25 (10)	1 (1)	24 (15)	
Other	15 (6)	5 (5)	10 (6)	
Prefer not to say	3 (1)	0	3 (2)	
**Employment or education, n (%)**				1 (1)
Working full-time (≥35 h/week)	175 (66)	71 (68)	104 (65)	
Working part-time (<35 h/week)	41 (16)	13 (12)	28 (18)	
Others (self-employed/casual)	5 (2)	4 (4)	10 (0.6)	
Unpaid work/parental duties	7 (3)	5 (5)	2 (1)	
Unemployed/unable to work	24 (9)	10 (10)	14 (9)	
Apprenticeship/student	11 (4)	1 (1)	10 (6)	
**Ethnicity, n (%)**				0
Australian	214 (81)	84 (80)	130 (82)	
Aboriginal/Torres Strait Islander	9 (3)	6 (6)	3 (2)	
New Zealander	22 (8)	10 (10)	12 (8)	
**Born in Australia, n (%)**	211 (80)	87 (83)	124 (78)	0
**Equivalized household income ^2^ (AUD), n (%)**				5 (2)
AUD 0–AUD 24,400	64 (24)	21 (20)	43 (27)	
AUD 24,401–AUD 37,100	31 (12)	11 (11)	20 (13)	
AUD 37,101–AUD 60,000	109 (41)	50 (48)	59 (37)	
AUD 60,001+	55 (21)	21 (20)	34 (21)	
**Source of income, n (%)**				0
Salary or wages	235 (89)	93 (89)	142 (89)	
Government support/pension	26 (10)	9 (9)	17 (11)	
Other	3 (1)	3 (2)	0	
**Have a healthcare card, n (%)**	79 (30)	28 (27)	51 (32)	1 (0.4)
**State or Territory, n (%)**				
Queensland	135 (51)	49 (47)	86 (54)	3 (1)
New South Wales	43 (16)	18 (17)	25 (16)	
Victoria	39 (15)	18 (17)	21 (13)	
Other (SA, WA, TAS, NT)	35 (13)	13 (12)	22 (14)	
Prefer not to say	9 (3)	5 (5)	4 (30)	
**SEIFA ISRD decile category ^3^, n (%)**				14 (5)
Low (1–3)	58 (22)	23 (22)	35 (22)	
Medium (4–6)	78 (30)	31 (30)	47 (30)	
High (7–10)	114 (43)	43 (41)	71 (45)	
**Housing types, n (%)**				5 (2)
House/townhouse	226(86)	90 (86)	136 (86)	
Apartment/flat	26 (10)	11 (11)	15 (9)	
Other	4 (2)	3 (3)	1 (1)	
**Frequency of moving residence in past 12 months**				1 (1)
None	176 (67)	62 (59)	114 (72)	
One or more	87 (33)	42 (40)	45 (28)	
**Parental stress, n (%)**				44 (17)
Unmanaged stress	146 (55)	60 (57)	86 (54)	
Well-managed stress	74 (28)	27 (26)	47 (30)	
**Number of adults**				0
One	43 (16)	12 (11)	31 (20)	
Two	203 (77)	90 (86)	113 (71)	
Three or more	18 (7)	3 (3)	15 (9)	
**Number of children (0–14 years)**				0
One	104 (39)	68 (65)	36 (23)	
Two	106 (40)	24 (23)	82 (52)	
Three or more	54 (21)	13 (12)	41 (26)	
**Number of children (15–17 years)**				0
None	260 (98.5)	104 (99)	156 (98)	
One or more	4 (1.5)	1 (1)	3 (2)	
** *Index child* **				
**Age (months),** **median (IQR)**	28.5 (15.3–47.6)	13.2 (9.2–18.3)	42.1 (33.5–53.6)	1 (1)
**Child gender, n (%)**				2 (0.8)
Boy	154 (58)	65 (62)	89 (56)	
Girl	104 (39)	36 (34)	68 (43)	
**Relationship to child, n (%)**				0
Biological father	262 (99)	104 (99)	158 (99)	
Stepfather	1 (0.5)	0	1 (1)	
Great grandfather	1 (0.5)	1 (1)	0	
**Days living with child per fortnight**				1 (1)
14 (full-time)	231 (88)	103 (98)	128 (81)	
7–13 (>50% of time)	11 (4)	0	11 (7)	
2–6 (<50% of time)	21 (8)	2 (2)	19 (12)	
**Attending childcare, n (%)**	163 (62)	52 (50)	111 (70)	1 (1)

IQR—interquartile range; BMI—body mass index; SEIFA—socioeconomic indexes for areas; ISRD—index of relative socioeconomic disadvantage. ^1^ Calculated from participant reported weight and height. ^2^ The mid-report of the household income bracket was divided by an equivalence factor (1 point to the first adult, 0.5 point to each additional person over 15 years old, 0.3 to each child under the age of 15). Equivalized household incomes were categorized into quartiles [39]. ^3^ Derived from participants’ postcode, with SEIFA 1 indicating the most disadvantaged [40].

**Table 2 nutrients-16-00205-t002:** Breastfeeding and paternal involvement in the total sample and for younger and older child groups.

	Total (n = 264)	Younger Child Group (<2 Years) (n = 105)	Older Child Group (2–5 Years) (n = 159)
**Breastfeeding, n (%)**			
Breastfeeding/ever breastfed	230 (87)	92 (88)	138 (87)
Never been breastfed	32 (12)	12 (11)	20 (13)
Missing	2 (1)	1 (1)	1 (1)
**Influence on child’s nutrition, n (%)**			
No influence	3 (1)	3 (3)	0
Some influence	109 (41)	45 (43)	64 (40)
A great deal of influence	150 (57)	55 (52)	94 (59)
Missing	2 (1)	1 (1)	1 (1)
**Preparing meals, n (%)**			
Not at all	4 (2)	4 (4)	0
Rarely	5 (2)	4 (4)	1 (1)
A few times a month	11 (4)	4 (4)	7 (4)
A few times a week	67 (25)	26 (25)	41 (26)
At least once a day	84 (32)	26 (25)	51 (32)
More than once a day	91 (35)	33 (31)	58 (37)
Missing	2 (1)	1 (1)	1 (1)
**Assisting child with eating, n (%)**			
Not at all	5 (2)	1 (1)	4 (3)
Rarely	10 (4)	0	10 (6)
A few times a month	10 (4)	3 (3)	7 (4)
A few times a week	53 (20)	14 (13)	39 (25)
At least once a day	104 (39)	41 (39)	63 (40)
More than once a day	79 (30)	44 (42)	35 (22)
Missing	3 (1)	2 (2)	1 (1)

**Table 3 nutrients-16-00205-t003:** Food security status as assessed by 18-item HFSSM and NHS single item.

Prevalence of Household Food Insecurity, n (%), (n = 222) ^1^				
	Food secure		Food insecure	
NHS–1 item ^2^	141 (63)		81 (37)	
HFSSM (dichotomized) ^3^	52 (23)		170 (77)	
HFSSM (categories)	High food security	Marginal food security	Low food security	Very low food security
Household level ^4^	22 (10)	30 (14)	72 (32)	98 (44)
Among adults ^5^	27 (12)	37 (17)	49 (22)	109 (49)
	High and marginal food security		Low food security	Very low food security
Among children ^6^	115 (52)		103 (46)	4 (2)

NHS—National Health Survey; HFSSM—household food security survey module. ^1^ Missing data for the food security measure (n = 42; 16%). ^2^ The single item asked: ‘In the last 12 months, was there any time you have run out of food and not been able to purchase more?’ Affirmative responses (yes) were categorized as food insecure. ^3^ Classified based on HFSSM guide; households with high or marginal food security were classified as food secure; low or very low food security was classified as food insecure. ^4^ Specification of food security was based on HFSSM (18 items) raw score: high food security (0), marginal food security (1–2), low food security (3–7), or very low food security (8–18) [36]. ^5^ HFSSM adult scale (10 items) was used to classify food security among adults. ^6^ HFSSM children scale (8 items) was used to classify food security among children.

**Table 4 nutrients-16-00205-t004:** Household and work chaos scores by food security status.

		Severity of Food Insecurity				Test Statistics (*p*-Value) ^2^
		High Food Security	Marginal Food Security	Low Food Security	Very Low Food Security	
CHAOS score ^1^, M ± SD	Total (n = 220)	14.38 ± 3.93	14.97 ± 4.68	15.69 ± 4.15	16.95 ± 4.56	3.13 (0.03 *)
	Younger child group ^3^ (n = 87)	15.00 ± 3.78	15.18 ± 4.09	15.60 ± 4.85	15.84 ± 4.39	0.12 (0.95)
	Older child group ^3^ (n = 133)	13.92 ± 0.69	14.83 ± 5.11	15.76 ± 3.62	17.62 ± 4.57	4.01 (0.01 *)
Work chaos score ^1^, M ± SD	Total (n = 214)	1.10 ± 1.04	1.21 ± 1.32	1.21 ± 1.24	1.37 ± 1.39	0.39 (0.76)
	Younger child group ^3^ (n = 85)	1.11 ± 1.17	1.55 ± 1.21	1.28 ± 1.36	1.25 ± 1.38	0.20 (0.90)
	Older child group ^3^ (n = 133)	1.08 ± 0.10	1.00 ± 1.37	1.14 ± 1.16	1.38 ± 1.40	0.57 (0.64)

M ± SD—mean ± standard deviation; CHAOS—Confusion, Hubbub, and Order Scale. ^1^ Missing data for household chaos (n = 44, 17%) and work chaos (n = 50, 19%). ^2^ One-way ANOVA was used to test for statistical difference. ^3^ Younger child group (child aged < 2 years); older child group (child aged 2 to 5 years). * *p* < 0.05.

**Table 5 nutrients-16-00205-t005:** Multiple linear regression models examining associations between paternal feeding practices, household food security, household chaos, and work chaos (final models).

Final Regression Model							
Feeding Practices		Independent Variables	B	95%CI	*p*-Value	Adjusted R^2^	ANOVA
Younger child group (<2 years) ^1^							
Coercive control	Using food to calm (n = 81)	Food insecure ^2^	0.122	−0.275, 0.519	0.542	0.156	0.003 **
		Household chaos	0.068	0.03, 0.105	<0.001 ***		
		Work chaos	−0.018	−0.145, 0.109	0.780		
		Child’s sex (boy) ^3^	0.308	−0.04, 0.656	0.082		
		Equivalized household income	0.000	0.000, 0.000	0.186		
	Persuasive feeding (n = 83)	Food insecure ^2^	0.552	0.106, 0.998	0.016 *	0.142	0.005 **
		Household chaos	0.056	0.011, 0.100	0.014 *		
		Work chaos	−0.008	−0.155, 0.139	0.917		
		Child’s age (months)	0.026	−0.008, 0.061	0.134		
		Father’s age	0.022	−0.008, 0.053	0.148		
	Parent-led feeding (n = 85)	Food insecure ^2^	0.472	0.048, 0.897	0.030 *	0.066	0.063
		Household chaos	0.033	−0.009, 0.074	0.119		
		Work chaos	0.022	−0.012, 0.164	0.761		
		Education ^4^	0.471	0.096, 0.845	0.014 *		
		Unmanaged stress ^5^	−0.228	−0.641, −0.186	0.277		
Structure	Family meal environment (n = 48)	Food insecure ^2^	−0.064	−0.628, 0.499	0.819	0.077	0.138
		Household chaos	−0.044	−0.1, 0.012	0.118		
		Work chaos	−0.082	−0.267, 0.102	0.373		
		Child’s sex (boy) ^3^	−0.570	−1.12, −0.19	0.043 *		
		Equivalized household income	−0.000	0.000, 0.000	0.100		
Older child group (2–5 years) ^1^							
Coercive Control	Reward for behavior (n = 132)	Food insecure ^2^	0.102	−0.217, 0.421	0.527	0.141	<0.001 ***
		Household chaos	0.035	0.004, 0.066	0.025*		
		Work chaos	0.075	−0.028, 0.178	0.152		
		Unmanaged stress ^5^	0.195	−0.088, 0.478	0.175		
		Father’s age	−0.034	−0.059, −0.01	0.006 **		
		Child’s age (months)	0.009	−0.001, 0.02	0.065		
	Reward for eating (n = 132)	Food insecure ^2^	−0.069	−0.437, 0.298	0.710	0.15	<0.001 ***
		Household chaos	0.038	0.003, 0.073	0.033 *		
		Work chaos	−0.001	−0.012, 0.118	0.988		
		Residential move (≥1) ^6^	0.443	0.108, 0.778	0.010 *		
		Child’s age (months)	0.019	0.007, 0.031	0.002 **		
		Father’s age	−0.023	−0.051, 0.005	0.107		
	Persuasive feeding (n = 131)	Food insecure ^2^	0.227	−0.014, 0.469	0.065	0.14	<0.001 ***
		Household chaos	0.015	−0.008, 0.037	0.198		
		Work chaos	0.012	−0.066, 0.089	0.762		
		Residential move (≥1) ^6^	0.309	0.094, 0.524	0.005 **		
		Father’s age	−0.019	−0.001, −0.194	0.043 *		
		Education ^4^	0.257	0.049, 0.464	0.016 *		
	Overt restriction (n = 133)	Food insecure ^2^	−0.175	−0.524, 0.175	0.324	0.09	<0.004 **
		Household chaos	0.048	0.014, 0.081	0.005 **		
		Work chaos	0.035	−0.078, −0.147	0.541		
		BMI ≥ 25 ^7^	−0.428	−0.752, −0.105	0.01 *		
		Residential move (≥1) ^6^	0.274	−0.044, 0.592	0.091		
Structure	Covert restriction (n = 133)	Food insecure ^2^	0.095	−0.279, 0.470	0.615	−0.011	0.666
		Household chaos	−0.021	−0.056, 0.014	0.242		
		Work chaos	0.001	−0.133, 0.108	0.837		
	Structured meal setting (n = 133)	Food insecure ^2^	−0.123	−0.486, 0.239	0.502	−0.005	0.500
		Household chaos	−0.021	−0.055, 0.014	0.236		
		Work chaos	0.001	−0.116, 0.117	0.987		
Autonomy Support	Offer new foods (n = 132)	Food insecure ^2^	−0.096	−0.359, 0.167	0.471	0.062	0.016 *
		Household chaos	−0.026	−0.051, −0.001	0.039 *		
		Work chaos	−0.036	−0.12, 0.048	0.401		
		Education ^4^	−0.287	−0.515, −0.058	0.014 *		
	Exploration of new foods (n = 132)	Food insecure ^2^	−0.146	−0.511, 0.219	0.430	0.053	0.035 *
		Household chaos	−0.013	−0.048, 0.021	0.442		
		Work chaos	−0.028	−0.145, 0.088	0.632		
		Education ^4^	−0.418	−0.739, −0.098	0.011 *		
		Child’s age	−0.014	−0.025, −0.002	0.021 *		
	Repeated presentation of new foods (n = 132)	Food insecure ^2^	−0.116	−0.429, 0.198	0.466	0.086	0.006 **
		Household chaos	−0.038	−0.067, −0.008	0.013 *		
		Work chaos	0.024	−0.076, 0.124	0.636		
		Education ^4^	−0.287	−0.562, −0.012	0.041 *		
		Child’s age	−0.012	−0.022, −0.002	0.017 *		

B—unstandardized coefficients; CI—confidence interval; R^2^—coefficient of determination; BMI—body mass index. ^1^ Younger child group completed FPSQ-S; older child group completed FPSQ-28 and FPI. Coding for categorical variables (where 0 is the reference group): ^2^ Food insecure = 1, food secure = 0. ^3^ Child’s sex: boy = 1, girl = 0. ^4^ Education: university degree or higher = 1, non-degree education = 0. ^5^ Unmanaged stress = 1, well-managed stress = 0. ^6^ Moved residence more than once in past 12 months = 1, not moved = 0. ^7^ BMI ≥ 25 = 1, BMI < 25 = 0. * = *p* < 0.05, ** = *p* <0.01, *** = *p* < 0.001.

## Data Availability

The data presented in this study are available on request from the corresponding author. The data are not publicly available due to privacy.

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
