# Peer review of "Dads at Mealtimes: Associations between Food Security, Household and Work Chaos, and Paternal Feeding Practices among Australian Fathers Living with Disadvantage"

_nutrients, 2024, doi:10.3390/nu16020205_

Round 1
Reviewer 1 Report
Comments and Suggestions for Authors
This is a very well written manuscript of a cross-sectional study examining the relationship between household chaos and feeding practices among low-income fathers. It was unique esp because of the focus on fathers. I have very few and minor comments.
1. The opening paragraph introduces the Nurturing Care Framework and is not discussed again. It set the stage for a different topic. I suggest taking that out and use a directly relevant introductory statement.
2. Just a comment - I was surprised by the substantial difference between the NHS and HFSSM.
3. Finally, add the study design as a limitation - it is not possible to determine directionality of relationships.
Reviewer 2 Report
Comments and Suggestions for Authors
Excellent approach to evaluate paternal involvement and practices on infant and children feedings among Australian fathres living with disvantage. The paper provide some clues to help those families to go across child feeding in a more positive way.
As in more than 3/4 of the sample there two adultsin the houshold It would be ideal to have the spouse answered the same questiosn as dads in order to evlauate the conistency of findings.
On Ref 15, how to access to it?
Reviewer 3 Report
Comments and Suggestions for Authors
Thank you for your paper, This article is highly valuable as it provides insights into the level of paternal involvement in children's lives, while also highlighting the challenges faced. Your paper is well written, and interesting. However, I do suggest some changes and points to address below:
Introduction and Background: The quoted passage asserts the importance of good nutrition for children, yet there is a lack of information regarding the current developmental status of children in Australia. It is suggested that the author incorporate data on the nutritional and developmental conditions of a sample of Australian children at various stages. Only by understanding this context can we appreciate the significance of this study. Otherwise, one might assume that Australian children are faring well, and the issue of paternal involvement may not be as significant.
Figure 1 depicts the study procedure and sample; hence, it is recommended to be placed in the second section.
Based on prior research, preterm birth should be a crucial indicator for child health. It is recommended that the author includes this as a control variable.
Reviewer 4 Report
Comments and Suggestions for Authors
Its is a positive sign to see someone writing more about fathers in childcare research, which is why I would firstly like to praise the authors for. Secondly there are a few minor issues, such as double spacing in the introduction and later on in other parts of the text. Thus, I suggest to grammatically check the manuscript once more
